# Blood Compatibility—An Important but Often Forgotten Aspect of the Characterization of Antimicrobial Peptides for Clinical Application

**DOI:** 10.3390/ijms20215426

**Published:** 2019-10-31

**Authors:** Stephan Harm, Karl Lohner, Ute Fichtinger, Claudia Schildböck, Jennifer Zottl, Jens Hartmann

**Affiliations:** 1Department for Biomedical Research, Danube University Krems, 3500 Krems, Austriaclaudia.schildboeck@donau-uni.ac.at (C.S.); jenn@live.at (J.Z.); jens.hartmann@donau-uni.ac.at (J.H.); 2Department of Pharmaceutical Technology and Biopharmaceutics, University of Vienna, 1090 Vienna, Austria; 3Institute of Molecular Biosciences, University Graz, 8010 Graz, Austria; karl.lohner@uni-graz.at

**Keywords:** antimicrobial peptides, inflammation, sepsis, endotoxin, blood compatibility

## Abstract

Acylation of antimicrobial peptides mimics the structure of the natural lipopeptide polymyxin B, and increases antimicrobial and endotoxin-neutralizing activities. In this study, the antimicrobial properties of lactoferrin-based LF11 peptides as well as blood compatibility as a function of acyl chain length were investigated. Beyond the classical hemolysis test, the biocompatibility was determined with human leukocytes and platelets, and the influence of antimicrobial peptides (AMPs) on the plasmatic coagulation and the complement system was investigated. The results of this study show that the acylation of cationic peptides significantly reduces blood tolerance. With increasing acyl chain length, the cytotoxicity of LF11 peptides to human blood cells also increased. This study also shows that acylated cationic antimicrobial peptides are inactivated by the presence of heparin. In addition, it could be shown that the immobilization of LF11 peptides leads to a loss of their antimicrobial properties.

## 1. Introduction

In recent years, multidrug resistance (MDR) of bacteria has become a global health problem, and there is an increasing demand for new antimicrobial agents and antibiotics. One of the most promising approaches in antibiotic development involves antimicrobial peptides (AMPs). AMPs are cationic amphipathic peptides representing an ancient host defense mechanism and are ubiquitous in living organisms. Many of them have a high affinity to components of the cell membrane of Gram-negative and/or Gram-positive bacteria. Therefore, compared to many conventional antibiotics which impair the bacterial metabolism, AMPs are less prone to the development of bacterial resistance since they interact directly with the bacterial membrane, which is an evolutionary highly conserved structure, and is not able to perform major structural changes without losing its various vital functions. The interaction of AMPs with the bacterial membrane initiates a change in the membrane curvature and leads to a destabilization and permeabilization of the membrane, finally resulting in the loss of the barrier function [1,2]. In order to improve the antimicrobial activity of some AMPs, their hydrophobic properties were increased by targeted N-acylation. This led to increased antimicrobial activity against *Escherichia coli*, which correlates with the degree of permeability of bacterial membranes of these peptides [3,4]. Some AMPs have a specifically high affinity to lipopolysaccharides (LPS, endotoxins) from Gram-negative bacteria and are able to suppress an LPS-induced release of tumor necrosis factor alpha (TNF-α) and could protect mice from lethal endotoxemia [3,4]. Among these is Polymyxin B (PMB), a naturally occurring AMP produced by *Bacillus polymyxa*. Due to its high affinity to LPS, it is used for a hemoperfusion adsorbent called Toraymyxin, which consists of a polystyrene-based woven fiber with covalently immobilized PMB. The manufacturer claims that Toraymyxin is able to remove LPS from patients suffering from endotoxemia using extracorporeal perfusion of the adsorbent cartridge. However, it has been shown recently that Toraymyxin acts as a delivery system of PMB by releasing the non-covalently bound fraction into the patients’ blood stream, and LPS is neutralized by PMB, which results in a prevention of the activation of pattern recognition receptors. Also, other adsorbents intended for the removal of LPS do not adsorb LPS sufficiently in in vitro experiments [5]. Consequently, there is currently no membrane/adsorption based blood purification system available that is able to remove LPS from the patients’ blood.

AMPs based on lactoferrin (LF), a transferrin in mammals and some invertebrates that has endotoxin-neutralizing properties [6], have proven to be potential candidates for use in extracorporeal therapies. They showed promising results by neutralizing endotoxins in the mouse model as well as by reducing late-onset sepsis in low-birth-weight neonates [7]. However, there is still a lack of knowledge of the substantial properties of these AMPs in order to facilitate their use in extracorporeal blood purification. In this study, LF11-based peptides derived from the pepsin cleavage product of human LF were characterized. Human LF is a multifunctional, iron-binding glycoprotein, which occurs in mammalian exocrine secretions and neutrophil granules. It has antimicrobial and LPS-binding properties [8,9]. Bovine lactoferricin (LFcin) has an α-helical structure, which is lost in an isolated form [10,11]. Human lactoferricin (hLFcin) comprises amino acid residues 1-45 of the N-terminus of human lactoferrin (hLF) and has increased antibacterial activity compared to intact hLF [12]. LFcin has animicrobial properties against Gram-positive bacteria, Gram-negative bacteria, and filamentous yeasts, including some antibiotic-resistant pathogens [13]. Wakabayashi et al. [13,14] and Strom et al. [15] were able to show that N-acylation of lactoferrin-based peptides could increase their antimicrobial activity. However, this usually leads to higher toxicity on eukaryotic cells, leading to loss of target cell selectivity [16,17].

The aim of this in vitro study was to characterize lactoferrin-based AMPs for their potential use in extracorporeal therapies with special attention to their biocompatibility, their binding affinity to LPS, and the capability to suppress cytokine release. Furthermore, it was examined whether the strongly polyanionic heparin has an influence on the antimicrobial effect of cationic AMPs. Selected AMPs were covalently immobilized on sepharose beads, and the resulting AMP-immobilized adsorbents were characterized for their antimicrobial properties. For this study, two synthetic LF11-based AMPs were selected, which showed good antimicrobial and LPS-binding properties in previous studies [18,19].

## 2. Results

### 2.1. Endotoxin Binding Affinity in an Aqueous Environment

The binding affinity of PMB to LPS was set to 100% and compared to the affinities of the various AMPs. All tested AMPs showed an affinity to LPS in an aqueous environment (Figure 1). From the LF11-322 peptides, the non-acylated and C_8_-C_12_ substituted peptides showed a binding affinity between 75% and 102% compared to PMB. Only the C_14_ acylated peptide loses its affinity to LPS, which is only 22%–26%, due to the long acyl chain. The reason for this could be that this strongly hydrophobic peptide forms stable micelle-like structures in an aqueous environment. This would result in almost no monomers being available for the LPS binding. From the LF11-227 peptides, the C_10_-substituted showed a significant increase in affinity to LPS (92%–109%). The other LF11-227 peptides showed a similar LPS affinity between 61% and 82%. There was no big difference in the affinity of the peptides to the different LPS molecules (*E. coli* vs. *P. aeruginosa*).

### 2.2. Endotoxin Binding Affinity in Human Plasma

Polymyxin B is known to be able to strongly reduce endotoxin activity by binding to LPS through the formation of the LPS-PMB complex in blood [20]. The LAL test was used to show whether synthetic peptides, which show a high affinity to LPS in aqueous solution, also possess this ability in human plasma. Therefore, 500 ng/mL AMP, LL-37 and PMB were added to serum spiked with 5 ng/mL LPS of *E. coli* and *P. aeruginosa*. Endotoxin activity was determined by the LAL test. Several AMPs show a binding affinity as high as PMB in an aqueous solution (Figure 1), but PMB outperforms all AMPs regarding the reduction of LAL activity in human serum (Figure 2). PMB was able to reduce the endotoxin activity of *E. coli* LPS by 63 ± 7% and of *P. aeruginosa* LPS by 74 ± 7%. The AMPs were only able to reduce the endotoxin activity of LPS derived from *P. aeruginosa*. A significant (*p* < 0.05) reduction of the endotoxin activity in serum was observed by the C_12_ and C_14_ acylated LF11-227 peptides and by the C_8_-C_12_ acylated LF11-322 peptides. The highest reduction of the endotoxin activity was caused by LF11-227-C_14_ (37 ± 5%).

### 2.3. Inhibition of Cytokine Release

Past studies [5,20] have shown that a PMB serum level between 100 to 200 ng reduces the cytokine secretion of LPS-stimulated leukocytes up to 90% in whole blood. PMB directly interacts with the conserved lipid A region of the LPS molecule. In vitro experiments showed that the formed LPS–PMB complex in blood exhibits lower inflammatory activity than the unbound LPS. In this study, it was tested whether the lactoferrin based peptides are able to form a complex with LPS similar to PMB, which would lead to a reduction of the inflammatory effect of LPS. The results (Figure 3) indicate that in contrast to PMB, none of the AMPs could reduce the inflammatory effect of LPS on leukocytes.

### 2.4. Minimum Inhibitory Concentration (MIC)

The MIC for the tested bacteria of the LF11-322 AMPs in aqueous solution (PBS) is between 4 and 64 μg/mL. It is noticeable that the MIC decreases with increasing acylation length of the peptide (Table 1). The antimicrobial effect of the LF11-322 peptides against *E. coli* and *S. aureus* decreased when the MIC was tested in 4% HSA solution. This is probably due to the partial binding of AMPs to albumin. In the LF11-227 peptides, only the non-acylated peptide showed antimicrobial properties in PBS, where the MIC was 8 μg/mL for both tested bacteria strains. Surprisingly, an antimicrobial effect of the acylated LF11-227 peptides was observed when the MIC was tested in 4 % HSA solution. For *S. aureus,* the MIC was 16 μg/mL and for *E. coli* the MIC was 64 μg/mL (Table 1). An explanation for this observation could be that in aqueous solution, the acylated LF-227 peptides form micellar like structures. If these hydrophobic peptides are solved in 4% HSA solution where the environment is more hydrophobic, then the amount of free antimicrobial peptides increases. In order to determine whether the antimicrobial properties of the peptides are influenced by the anionic anticoagulant heparin, the MIC was additionally tested in PBS containing 5 IU/mL heparin. It was clearly shown that the antimicrobial effect of the peptides is eliminated by the presence of heparin. The reason for this observation might be that the polyanionic heparin binds the cationic peptides and thus neutralizes their antimicrobial effect. The conventional antibiotic ciprofloxacin showed a MIC of <0.0625 μg/mL in all tested solutions. Ciprofloxacin was used as a control antibiotic because it has a broad spectrum of action at very low dosage. Ciprofloxacin has an MIC_90_ of <1 mg/L against all species of *Enterobacteriaceae* and *Staphylococcus* [21]. In past studies, no mutual influences between ciprofloxacin and heparin could be observed [22]. The MIC of PMB for *E. coli* in PBS was 2 μg/mL, in 4% HSA solution it was 4 μg/mL, and in heparin spiked PBS was also 4 μg/mL. In contrast, the MIC of PMB against *S. aureus* was significantly higher and completely lost when heparin was added. LL-37 showed antimicrobial properties only against *E. coli*, which were also eliminated by heparin (Table 1).

### 2.5. Interaction of AMPs with the Outer Membrane

To determine whether the AMPs can permeabilize the outer membrane of Gram-negative bacteria, the AMP induced 1-N-Phenylnaphtylamine (NPN) uptake was determined. NPN is a hydrophobic fluorescent dye whose fluorescence emission is improved in an environment of glycerophospholipid [23]. A rising fluorescence signal therefore means a destabilization, destruction, or loss of the outer membrane of Gram-negative bacteria. Overall, the NPN uptake study shows that N-acylated peptides interact more strongly with the outer membrane of *E. coli* than non-acylated peptides (Figure 4). The highest fluorescence increase was achieved by the use of C_12_ acylated peptides, suggesting that these peptides cause the strongest destabilization to the outer membrane. This indicates that AMPs with longer lipid lengths interact more strongly with the outer membrane of *E. coli* but has its optimum with C_12_ acylation. The use of C_14_ acylated peptides resulted in a lower NPN uptake compared to C_12_ acylated peptides.

### 2.6. Scanning Electron Microscopy (SEM)

SEM was used to visualize the cell wall damage of bacteria cells. While no damage of bacteria was observed in the untreated control, the AMP caused lysis of Gram-negative as well as Gram-positive cells (Figure 5). PMB, an antimicrobial peptide which permeabilizes the outer membrane of Gram-negative bacteria by LPS binding, was not able to lyse *S. aureus* cells. This illustrates that in comparison to PMB, the tested AMPs do not have a selective antimicrobial effect. Instead, they unselectively destroy all bacteria membranes. Of course, it cannot be ruled out that blood or tissue cells may also be attacked by these AMPs. To verify this more accurately, additional blood compatibility tests were carried out.

### 2.7. Hemolysis

The hemolysis test was performed with fresh human blood cells, RBC suspended in PBS and whole blood were spiked with different AMP concentrations and hemolysis was measured with the Kahn method. The results clearly show that the hemolytic activity in aqueous medium (PBS) is significantly higher than in whole blood (Table 2). Compared to the control (blood without additives), the non-acylated peptides did not cause hemolysis. In PBS the C_14_ acylated peptide of the LF11-322 peptide and the C_12_ acylated peptide of the LF-227 peptide showed the highest hemolytic activity. Furthermore, the results show that the hemolytic effect is proportional to the concentration used. In whole blood, where plasma proteins are present, none of the tested peptides caused hemolysis. The explanation for this observation could be that the plasma proteins partially bind the AMPs which leads to a reduction of their hemolytic activity. This would explain the different MIC values of AMPs in PBS and HSA solution (Table 2) and shows how important it is to check antimicrobial properties and biocompatibility tests in the medium in which the AMPs are intended to be used.

### 2.8. Plasmatic Clotting

The results show that the conventional antibiotics LL-37 and PMB do not affect the plasmatic coagulation of heparinized and citrate containing plasma (Table 3). In heparinized plasma, coagulation was activated by all acylated variants of the two peptides. The non-acylated peptides do not cause plasma clotting. When citrate anticoagulated plasma was used, only the acylated LF11-322 peptides triggered the plasmatic coagulation cascade.

Since the peptides of the LF11-322 family can even activate coagulation in citrate plasma, this was investigated more closely. The results show that only the acylated LF11-322 peptides cause coagulation of citrate plasma (Table 4). Concentrations as low as 32 μg/mL of the C_14_ acylated peptide trigger clotting of citrate plasma within one hour. The peptide with the shortest acyl chain (C_8_) caused coagulation of citrate plasma from 256 μg/mL upwards. These measurements show that the activation of plasmatic clotting depends on the peptide concentration as well as on the degree of acylation.

### 2.9. Platelet Activation

To investigate the influence of the peptides on platelets, separate batch tests were performed. For this purpose, 128 μg/mL of AMP, PMB, and LL-37 were incubated in platelets rich plasma (PRP) for 2 h. The results show that the C_12_ and C_14_ acylated LF11-322 peptides significantly reduce the platelet count in citrated plasma (Figure 6). This strongly suggests an activation of the thrombocytes by the acylated LF11-322 peptides. For these peptides, no significant difference in platelet count was observed in heparinized PRP. A possible reason for this could be that the polyanionic heparin molecule inactivates the positively charged AMPs by building a polyelectrolyte complex. With LF11-227 peptides, no significant difference in the number of platelets could be measured in citrate anticoagulated PRP. However, PRP anticoagulated with heparin showed a strong turbidity when acylated LF11-227 peptides were added. This made it impossible to count the platelets using the Sysmex Automated Hematology Analyzer. The same observation was made with LL-37 in heparinized PRP. In order to investigate the cause of the turbidity, samples for electron microscopy were prepared.

To check the platelet morphology, microscopic visualization was performed using a scanning electron microscope. Based on cell morphology, Goodmann [25] categorized the platelets into five morphological forms ranked according to the increasing stage of their activation. These are, in order of the degree of activation, discoid (round), dentritic, spreading dendritic, spreaded and fully spreaded platelets. The electron microscopic images (Figure 7d) clearly show that when citrate anticoagulated PRP is incubated with the 322-C_14_ peptide, there are hardly any intact platelets left. This peptide destroys the membrane of the thrombocytes through its strongly hydrophobic acyl group, whereby only cell fragments (microvesicles) are visible in the SEM specimen. The PRP samples without AMP (Figure 7a) or with 128 μg/mL PMB (Figure 7b) contain morphologically intact platelets. When platelets are incubated with the non-acylated 322 peptide, they strongly change their shape (Figure 7c). There are hardly any round platelets visible in the SEM preparations. The platelets are strongly activated and thus are morphologically strongly spreading or adherent. Incubation of the acylated LF11-227 peptides (Figure 7e) in heparin anticoagulated PRP showed precipitates caused by plasmatic coagulation. The SEM images show that only few morphologically round platelets are present. Most platelets are strongly adherent or destroyed. Many cell fragments (microvesicles) are visible, indicating lysis of the platelets. Microvesicles have a high density of phospatidylserine (PS) on the extracellular membrane side. The negatively charged PS groups are able to bind positively charged AMPs, which in turn can be cross-linked by ionic interaction of the heparin polymer. This cross-linking could be a possible cause of turbidity of AMP containing heparinized PRP.

### 2.10. Interaction with Leukocytes (Flow Cytometry)

In order to determine whether the AMPs destabilize the membrane of leukocytes and thus destroy them, whole blood was spiked with 128 μg/mL AMP. After an incubation period of 120 min, the viability of the leukocytes was measured by flow cytometry. In addition to the AMP samples, the antibiotics ciprofloxacin and PMB were tested. Among the conventional antibiotics, only ciprofloxacin showed a cell-lysing effect on leukocytes (Figure 8). However, it should be noted that the serum level of ciprofloxacin during clinical treatment is about 3 μg/mL. The concentration used in this experiment was 128 μg/mL, which is 40 times higher than the recommended dose. However, the cytotoxic and cytostatic effects of ciprofloxacin on leukocytes, in particular on T cells, are known [26]. In the tested AMPs only the LF11-322 peptides showed strong cytotoxic effects on leukocytes in whole blood. As shown in Figure 8, the number of dead leukocytes increases with the acyl chain length of the peptide. The non-acylated peptide has no cytotoxic properties on leukocytes. The highest percentage of dead leukocytes was induced by the LF11-322-C_14_ peptide with 21 ± 16%.

### 2.11. Complement Activation

In order to check whether the AMPs activate the complement system in addition to coagulation, a separate batch test was performed. There are three different pathways of complement activation: the classical pathway, the alternative pathway, and the lectin pathway. In all three pathways, the active complement factor C3a is formed. Therefore, C3a was analyzed as an indicator of whether and to what extent the complement system is activated by the addition of AMPs. The results clearly show that the complement system of the LF11-322 peptide is activated only by acylated peptides (Figure 9). Similar to plasmatic coagulation, the activation of the complement system depends on the acyl chain length. The highest C3a level was achieved by LF11-322-C_14_. The other peptides tested did not increase C3a levels compared to the control.

### 2.12. Antimicrobial Properties of Immobilized AMPs

To investigate the antimicrobial properties of the previously characterized antimicrobial peptides in an immobilized state, they were immobilized on adsorbent surfaces. It was striking that the coupling efficiency was significantly higher for non-acylated AMPs and PMB than for acylated AMPs (Table 5). This could be due to the fact that the coupling of NHS-activated surface works more efficiently with terminal peptide amines, which are not present in the acylated AMPs. The growth of bacteria could not be prevented even by using 20 % (*v/v*) AMP or PMB immobilized adsorbent. Although the growth determined by turbidity measurement (OD 620 nm) was reduced, it was not inhibited. It could also be shown that growth medium, which was incubated with the AMP immobilized adsorbent for 2 h shortly before the MIC test was performed, also caused a reduction of bacterial growth. This indicates a release of immobilized but not covalently linked peptides (Figure 10) from the adsorbent surface.

## 3. Discussion

David et al. suggested that a clear separation of cationic and apolar domains (amphiphilic character) is crucial for the design of a molecule for LPS sequestration [27]. To neutralize endotoxins, the binding of peptides to the LPS molecule alone is not sufficient [28]. Studies from other groups have shown that for certain cationic AMPs, a fatty acid substitution of ≥ C_8_ is necessary for efficient LPS neutralization [29,30]. For this reason, only lipid acylations ≥ C_8_ were selected for this study. According to the results of the NPN permeability study, the optimal lipid chain length of LF11 peptides for antimicrobial properties is C_12_. This confirms the results of scientific studies carried out by other groups [31,32]. The results show that the necessary acyl chain length in order to maximize the binding affinity to LPS depends not only on the AMP, but also on the medium in which the experiment is carried out and the origin of LPS (*E. coli* vs *P. aeruginosa*). Some of the lipid substituted peptides were able to reduce significantly the endotoxin activity of *P. aeruginosa* in serum, but showed no effect against LPS from *E. coli*. This can be attributed to the fact that bacterial strains differ in the acylation (number, length) of the lipid A moiety of their endotoxins. In contrast, PMB is able to neutralize both LPS types. The cytokine release study confirms that the tested LF11 peptides are not able to neutralize the endotoxins in serum. Only PMB can form a stable LPS-PMB complex in whole blood and thus reduce the release of pro-inflammatory mediators of leukocytes [20].

When determining the MIC, it can be clearly seen that the antimicrobial effect of AMPs strongly depends on the medium to be tested. The antimicrobial properties of the LF11-322 peptides against *E. coli* and *S. aureus* are reduced in albumin solution, whereas the acylated LF11-227 peptides show only an antimicrobial effect in protein solution. In general, the MIC of *S. aureus* is slightly lower than that of *E. coli*. This suggests that unlike PMB, the antimicrobial effect is not based on LPS binding and that the peptides do not selectively act against a bacterial group. The polyanionic heparin is able to cancel the antimicrobial effect of the AMPs in our MIC study. A possible explanation for this could be that the anionic heparin binds the cationic AMPs and thus inactivates them. Previous works have shown interactions between heparin and natural cationic AMPs such as LL-37 and protamine [33,34]. These studies also assume an ionic interaction between the anionic heparin and the cationic peptides. Further and especially targeted studies would be necessary to clarify this observed effect. This fact must be taken into account for clinical applications of AMPs, especially for patients who are subject to extracorporeal blood purification therapy. Their blood must be sufficiently anticoagulated before and during treatment and the most frequently used anticoagulant in intensive care medicine is heparin. LL-37 is a human antimicrobial peptide from the group of cathelicidins. It is mainly produced in immune cells and is part of the innate immune response as well as the apoptosis of endogenous cells. In the MIC study, LL-37 shows an antimicrobial effect only against *E. coli*, which is also lost in the presence of heparin. PMB loses its antimicrobial effect induced by heparin only against Gram-positive bacteria. This shows that the affinity of PMB to LPS is not only selective, but almost specific and thus is significantly higher than to heparin. Although some synthetic AMPs have shown promising results in in vitro tests, especially in MIC determinations, none have yet been approved for clinical use [30,32,35]. The difficulty lies in synthesizing a peptide that possesses antimicrobial properties against a broad spectrum of multiresistant germs on the one hand and on the other hand has good blood compatibility. Toxicological tests with cell-lines cultured in culture media or hemolysis tests with erythrocytes are usually performed as biocompatibility tests [30,31,36,37]. However, several and more complex tests are required to test the blood compatibility of antibiotics. In addition to the hemolysis test, we also investigated the influence of the LF11 peptides on the coagulation and complement system. Furthermore, it was investigated whether the AMPs only attack the cell membrane of bacteria or whether they also attack the membrane of erythrocytes, leukocytes and thrombocytes. The determination of hemolysis is a marker for the selectivity of a membrane-destabilizing AMP. The profound difference between bacterial and eukaryotic cell membranes is the fact that bacterial membranes are negatively charged due to embedded negatively charged lipids, while membranes from eukaryotic cells are neutral. This fact should allow appropriate AMPs to destabilize selectively the bacterial membrane, while leaving the eukaryotic cell widely unaffected. Since for the destabilization process of bacterial membranes amphipathic AMPs are advantageous [38], only moderately acylated AMPs should be used in order to reduce the risk of destabilizing eukaryotic cells. In the presence of a membrane mimetic environment, the short (nine amino-acids) LF11-322 folds into a loop comprising a short helical segment [39], which separates the cationic and hydrophobic residues along the molecular axis of the peptide. The secondary structure prediction using the program PEP-FOLD indicated a similar structure of LF11-227 (http://bioserv.rpbs.univ-paris-diderot.fr/services/PEP-FOLD/). The structure of both peptides is in accordance with the structure of the parent peptide LF11. N-acylation forced the peptide chain to wrap around the acyl chain, resulting in an even better defined fold [40]. It can be expected that N-acylation of LF11-322 and LF11-227 follows the same principle. In our blood compatibility test we could show that with increasing acyl chain length of the LF11-322 peptide also leukocytes, RBC and platelets are lysed. The same tendency could be observed with the activation of the complement system and the plasmatic coagulation. In whole blood, where the blood cells are suspended in plasma, no hemolysis could be detected by acylated LF11 peptides. This indicates that most of these peptides are bound to plasma proteins. The SEM images confirm our results that PMB selectively destabilizes Gram-negative cell membranes, while Gram-positive bacteria and blood cells are not lysed or destabilized by PMB. In contrast, AMPs destabilize Gram-positive as well as Gram-negative cells. Together with the results of the blood compatibility tests, this suggests that depending on the concentration, acylated AMPs interact less selectively with all biological membranes, resulting in poor blood compatibility. In this study, the immobilization of the AMPs was conducted via their amino-groups. Since the AMPs used in this study are arginine-rich, it can be assumed that each AMP was immobilized via different amino-groups, leading to an undefined, varying orientation of the AMP on the adsorbents surface. However, the orientation of the AMP plays an important role in its efficiency [41], and the random orientation might lead to suboptimal adsorption characteristics of the adsorbent. Overall, it can be said that the antimicrobial property of AMPs and also of PMB is lost through immobilization on a surface. This can be explained by the fact that the peptides lose the ability to diffuse into the bacterial membrane to make it permeable due to immobilization on a surface.

## 4. Conclusions

The acylation of cationic LF11 peptides leads to an improvement of the antimicrobial properties, but at the same time causes a deterioration of blood compatibility. Some synthetic AMPs, especially when optimized for the destabilization of bacterial cell membranes, also destabilize or even lyse blood cells. Therefore, for applications with direct contact to blood cells, AMPs have to be chosen carefully in order to find the correct tradeoff between antimicrobial activity and biocompatibility.

## 5. Materials and Methods

### 5.1. Materials and Bacterial Strain

LPS from *E. coli* 055: B5 and *P. aeruginosa* 10 as well as PMB and the human host defense peptide LL-37 (LL-37 trifluoroacetate salt) were purchased from Sigma-Aldrich (St. Louis, MO, USA). Ciprofloxacin was purchased from Sigma-Aldrich (St. Louis, MO, USA). For the MIC study, bacterial strains from *E. coli* (ATCC 11303) and *S. aureus* (ATCC 12600) were purchased from American Type Culture Collection, Manassas, VA, USA. Cytokine quantification was carried out using a Magnetic Luminex Assay from R&D Systems (Minneapolis, MN, USA) on a Bio-plex-200 Analyzer (Bio-Rad Laboratories, Hercules, CA, USA).

### 5.2. Animicrobial Peptides

The AMPs used in this study are shown in Table 6. Different acylations were chosen in order to optimize their binding affinity to LPS as well as their capability to neutralize LPS. Peptides were purchased from PolyPeptide Laboratories (San Diego, CA, USA) and were synthesized using FMOC-chemistry. Purity of the peptides were >96% as determined by RP-HPLC and MS. The AMPs were obtained in lyophilized form and dissolved in 0.1 M 2-(*N*-Morpholino)ethansulfonacid (MES) buffer at a concentration of 1 mg/mL. The peptide solutions were prepared weekly and stored at 4 °C.

### 5.3. Blood Donation

The blood donations were approved by the ethics committee of Danube University Krems (21/01/2012) and were taken from healthy volunteers who had to sign an informed consent form. Tubes (Vacuettes) for blood donation were purchases from Greiner Holding AG (Kremsmünster, Austria). Plasma or serum was obtained by centrifuging blood or coagulated blood at 3500 *g* for 10 min.

### 5.4. Endotoxin Binding Affinity

The characterization of the affinity of AMPs to LPS from *E. coli* and *P. aeruginosa* was carried out by a fluorescent displacement assay based on BODIPY TR cadaverine (BC) [42] in microtiter plates. BC is a fluorescent reagent that tightly binds via ionic bonds to the lipid A moiety of LPS and forms a complex. Complex formation quenches the fluorescence of BC. When molecules that interact with LPS are added, BC will be displaced from the complex, with concomitant dequenching of its fluorescence emission. BC was purchased from ThermoFischer Scientific (Waltham, MA, USA). In this study, the assay solution consisted of 7.5 μg/mL LPS and 2.1 μM Bodipy TR cadaverine in TRIS buffer (50 mM, pH 7).

AMPs were added at concentrations ranging from 565 to 3387 ng/mL to the assay solution. After 30 min of incubation, fluorescence measurements were carried out using a Synergy 2 microplate reader (BioTek, VT, USA) at excitation and emission wavelengths of 580 and 620 nm, respectively. The LPS binding affinity was calculated from the area under the curve, which was drawn as fluorescence intensity vs. AMP concentration using Microsoft Excel for Windows 2016.

### 5.5. Endotoxin Inactivation

The LPS neutralization study in serum was conducted using a chromogenic Limulus Amebocyte Lysate (LAL) test (Charles River Laboratories, MA, USA). The LAL test is the most established method for quantifying LPS. This method uses amebocyte lysate from *Limulus polyphemus*, wherein the presence of LPS triggers a defense mechanism that induces coagulation. The LAL test is used successfully for detection of LPS in a wide variety of industrial, pharmaceutical, and research applications, such as water supplies, parenteral fluids, drugs for intravenous administration, and certain biologic fluids (e.g., cerebrospinal fluid). Therefore, serum, which was obtained from freshly drawn blood, was spiked with 500 ng/mL AMPs, PMB or LL-37 before adding 5 ng/mL LPS from *E. coli* or *P. aeruginosa*. As positive and negative controls, native serum with and without LPS was used, respectively. Endotoxin activity was measured using the LAL assay according to the manufacturer’s specifications.

### 5.6. Inhibition of Cytokine Release

By adding LPS to the blood, leukocytes are stimulated to produce inflammatory mediators (cytokines). In this experiment it was tested whether, similar to PMB, the activation of leukocytes by LPS can be reduced by AMP. The capability of AMPs to reduce or inhibit cytokine release by neutralizing endotoxins was evaluated in fresh heparinized (5 IU/mL) blood spiked with 150 ng/mL AMPs, PMB, or LL-37 followed by adding 0.5 ng/mL endotoxin from *E. coli*. Blood without any additive and blood spiked with endotoxins was used as the negative and positive control, respectively. After 4 h of incubation on a roller mixer at 37 °C, the plasma was separated by centrifugation and cytokine quantification (TNF-α, IL-1β, IL-6, and IL-8) was carried out.

### 5.7. Minimum Inhibitory Concentration (MIC)

The determination of the MIC was carried out in aqueous solution (0.01 M phosphate buffer, 0.0027 M potassium chloride and 0.137 M sodium chloride, pH 7.4) as well as in 4% human serum albumin (HSA; Kedrion Biopharmaceuticals, Gallicano, Italy) based on the standardized method described in DIN 58940-8 [43]. Additionally, the MIC testing was also performed in aqueous solution containing 5 IU/mL heparin. For bacterial growth, a Müller-Hinton broth (Carl Roth, Karlsruhe, Germany) was used. The inoculum with 2 × 10^5^–8 × 10^5^ colony forming units (CFU) per mL was spiked with AMPs, PMB, LL-37, and commercially available antibiotics at concentrations from 0.0625 to 128 μg/mL. After overnight incubation at 37 °C, the MIC was determined by turbidity measurement at 620 nm using an Anthos ht3 plate reader (Biochrom, Cambridge, UK).

### 5.8. 1-N-Phenylnaphtylamine (NPN) Uptake Assay

The uptake of the fluorescent dye 1-*N*-phenylnaphthylamine (NPN; Merck, Darmstadt, Germany) allows the measurement of permeability changes of the outer membrane of Gram-negative bacteria [23,44]. Hydrophobic substances such as NPN are excluded by an intact cell membrane. However, once damaged, the entry of NPN to the phospholipid layer is possible, resulting in prominent fluorescence. According to Koh et al. [30], a bacterial suspension of *E. coli* with an optical density of 0.4 at 600 nm (OD_600_) was incubated with 40 μM NPN dye in equal volumes. AMPs were added to a final concentration ranging from 8 to 64 μg/mL to the bacteria suspension followed by incubation for 1 h at 37 °C. After incubation, fluorescence was measured at excitation and emission wavelengths of 335 nm and 405 nm, respectively, using a Synergy2 microplate reader (BioTek, Winooski, VT, USA).

### 5.9. Scanning Electron Microscopy (SEM)

In order to verify and to visualize the impact of AMPs on the bacterial membrane, electron microscopy was conducted. Therefore, bacterial suspensions of *E. coli* and *S. aureus* with a concentration of 2 × 10^5^–8 × 10^5^ CFU/mL were incubated with 128 μg/mL of one AMP (LF11-322-C_8_) and PMB in a microplate over night at 37 °C. After overnight culture, the bacteria were prepared for SEM. To achieve this, the bacteria suspensions were filtered through a sterile filter (Swinnex filter holders, Merck Millipore, Burlington, MA, USA). The filters with the collected bacteria were than washed with PBS (10 mM, pH 7.2) and fixed with a 2.5% glutaraldehyde solution (Carl Roth, Karlsruhe, Deutschland). The fixed specimens were gradually dehydrated by increasing ethanol solutions (10% to 100%) and sputtered with gold (Q150R ES Sputter Coater, Quorum Technologies Ltd., East Sussex, UK) prior to examination with a scanning electron microscope (FlexSEM 1000, Hitachi, Tokyo, Japan).

### 5.10. Hemolysis

Hemolysis was tested with fresh citrate anticoagulated blood from healthy volunteers as well as with blood cells which were suspended in phosphate buffered saline (10 mM pH 7.4). Afterwards, 100 μL AMP solution was added to 500 μL blood to finally achieve AMP concentrations of 8, 32, and 128 μg/mL. As the negative and positive control, 100 μL PBS buffer and 100 μL 12% Triton-X buffer was added to 500 μL blood, respectively. After incubation on an orbital shaker for 60 min at 37 °C, the cells were removed by centrifugation and 100 μL of the supernatant were transferred to a 96 well microplate for hemolysis measurement. The hemolysis was calculated from the extinction at the wavelengths 562, 578, and 598 nm with a Synergy 2 microplate reader (BioTek, Winooski, VT, USA) according to the method published by Kahn [24].

### 5.11. Activation of the Clotting Cascade

To check the influence of AMPs on heparin and citrate anticoagulation, freshly heparinized (0.5 IU/mL) and citrated (17 mM) plasma was spiked with 128 μg/mL AMP, PMB, and LL-37. In addition, conventional antibiotics with a final plasma level of 128 μg/mL were included in the test. Plasma spiked with MES buffer without AMP was used as the negative control. Plasma with 128 μg/mL protamine, which is used as antidote for heparin and 2 μL calcium (500 mM)- magnesium (250 mM) chloride solution for citrate binding, was used as the positive control. Samples were incubated at 37 °C for 120 min. Coagulation was determined by turbidity measurement at 600 nm using a well plate reader.

### 5.12. Clotting Batch Test in Citrate Plasma

Since the acylated LF11-322 peptides cause plasmatic coagulation of citrate anticoagulated plasma, separate clotting experiments were performed with this peptide group. The aim was to check whether the coagulation activation is dependent on the AMP concentration and on the acyl chain length. For this purpose, plasma from freshly drawn citrated blood was spiked with different amounts (0, 8, 32, 128, and 256 μg/mL) of LF11-322 peptides, LL-37, and PMB. Plasma without additives served as the negative control, whereas plasma spiked with CaCl_2_/MgCl_2_ (500/250 mM) solution to restore the physiological calcium and magnesium level in plasma was used as the positive control. Incubation was performed at 37 °C for 60 min. Clot formation was determined by turbidity measurement at 600 nm using a plate reader. In order to ensure that the measured turbidity of the plasma is caused by fibrin formation, the fibrinogen and D-Dimer concentrations were additionally determined. During plasmatic clotting, the fibrinogen level decreases and the D-Dimer level in serum increases. Both were analyzed on a Sysmex CA560 with the corresponding reagents (Siemens Healthineers, Erlangen, Germany).

### 5.13. Platelets Activation

Since it was observed that some AMPs are able to activate the plasmatic coagulation system of heparinized and citrate anticoagulated blood, the influence of this peptides on platelets was investigated. For this purpose, fresh heparinized and citrated blood were collected from three donors. The blood was centrifuged at 500 g for 5 min and platelet rich plasma (PRP) was obtained. The PRP was incubated at 37 °C for 2 h with 128 μg/mL AMP, PMB, and LL-37. PRP without any additives served as the control. After incubation, the platelet number was determined by a Sysmex Automated Hematology Analyzer KX-21-N (Kobe, Japan). In addition, the PRP samples were prepared for SEM to visualize the morphology of the thrombocytes. After incubation, 10 μL PRP were diluted with 990 μL PBS (0.01 M) and filtered through a 0.2 μm membrane (Membrane Filter Isopore 0.2 μm, Merck, Darmstadt, Germany). The platelets were retained on the membrane, washed with 1 mL PBS, and fixed with 1 mL glutaraldehyde solution (2.5% glutaraldehyde in physiological saline solution). After an incubation time of 60 min, the samples were dehydrated using a series of ethanol (30%, 50%, 60%, 70%, 80%, 90%, and 100%). Afterwards the samples were dried overnight at RT, sputtered with gold and visualized with a scanning electron microscope (FLEXSem 1000, Hitachi, Japan).

### 5.14. Flow Cytometry

The measurement of the membrane destabilization of leukocytes by AMPs was carried out by flow cytometry. In order to determine the cytotoxic properties of the tested AMPs on human leukocytes, fresh citrate anticoagulated blood was mixed with 128 μg/mL AMP and incubated for 60 min at 37 °C. In addition, blood samples were incubated with 128 μg/mL PMB and ciprofloxacin. Whole blood without AMPs or antibiotics was used as a control. After incubation, the proportion of apoptotic and dead cells was determined by flow cytometry. To achieve this, blood samples were diluted 1:100 with 0.01 M phosphate buffered saline and centrifuged at 300 *g* for 10 min. The supernatant was removed, and cells were stained with 10 μL CD45 pacific blue (leukocyte marker; Beckman coulter, Brea, CA) and 10 μL 7-AAD (a marker for dead cells; Sigma-Aldrich, Steinheim, Germany) for 15 min at room temperature in the dark. After incubation, 1.8 mL lysis buffer (1:10 dilution in distilled water) was added, mixed, and incubated for 10 min at room temperature in the dark. This was followed by two washing steps (2300 *g*, 5 min, room temperature) and the cells were resuspended in a final volume of 200 μL 0.01 M PBS for measurement. Flow cytometry was performed using a Gallios flow cytometer (Beckman Coulter, Brea, CA, USA) equipped with 405 nm, 488 nm, and 638 nm lasers and the Kaluza Analysis Software (Beckman Coulter, Brea, CA, USA), which was used for measurement and data analysis. Blood cells which were incubated in 70% methanol before staining served as the positive control and blood cells without any additives were used as the negative control.

### 5.15. Complement Activation

The complement system is part of the innate immunity and consists of plasma proteins that can be activated in the course of the immune response on numerous foreign surfaces. Separate tests were performed to check whether the AMPs used in this study are able to activate the human complement system. Plasma from fresh donated blood, anticoagulated with citrate (5 mM), was spiked with 128 μg/mL AMP, PMB and LL-37 and incubated for 60 min at 37 °C on a roller mixer. Plasma spiked with 1 μg/mL LPS from *E. coli* served as the positive control and plasma without any additive was used as the negative control. After incubation, the complement system was stopped by adding 100 μL ice cold EDTA solution (10%, *w/v*) per mL plasma. Samples were stored at −80 °C until quantification of the complement factor C3a by ELISA (C3a des Arg ELISA kit, Abcam, Cambridge, UK) was performed.

### 5.16. Antimicrobial Properties of Immobilized AMPs

In order to determine whether the AMPs retain their antimicrobial property after immobilizing onto a surface, selected AMPs and PMB were covalently bound to agarose beads under sterile conditions. Therefore, the most hydrophilic and the most hydrophobic AMP from each peptide family were chosen, namely the non-acylated LF11-322 and LF11-227 and the C_14_-acylated M-LF11-322 and M-LF11-227 peptides. The immobilization was achieved by adding 400 μL peptide solution (1 mg/mL) on *N*-hydroxysuccinimid (NHS) activated agarose beads (Pierce spin columns, ThermoFisher Scientific, MA, USA) followed by two hours of incubation at room temperature in an overhead shaker. The activated agarose contains NHS ester functional groups that react with primary amines on AMPs to form stable amide linkages. The coupling reaction is performed in an amine-free buffer at pH 7.4 (PBS). After incubation, the peptide content of the supernatant was quantified (Pierce 660 nm Protein Assay, ThermoFisher Scientific, Waltham, MA, USA) in order to determine the coupling efficiency.

To check the antimicrobial properties, 200 μL bacteria suspensions of *E. coli* (2 × 10^5^–8 × 10^5^ CFU/mL) were incubated with 5%, 10%, and 20% (*v/v*) of the AMP-immobilized adsorbent in 96 microplate wells over night at 37 °C. After incubation, bacterial growth of the supernatant was determined by OD_620 nm_ measurement using an Anthos HT3 microplate reader. To determine whether the antimicrobial effect is caused by a peptide release from the adsorbent surface, additional release samples were performed according to the protocol of Rapsch et al. [37]. For this purpose, 20% (*v/v*) AMP adsorbent was incubated in Müller-Hinton broth for 2 h. The adsorbent was then centrifuged and the supernatant was incubated with the same bacteria suspension under the same conditions as described above. In addition, adsorbent with Tris deactivated surface was used as control adsorbent and the bacteria suspension without adsorbent was used as the positive control.

### 5.17. Statistics

All tests were carried out at least in triplicate. Calculations of standard deviations and area under the curve were carried out using Microsoft Excel 2010 (Microsoft, WA, USA). All other statistical calculations were carried out with SigmaStat for Windows 2.03. The Kolmogorov-Smirnov Test was applied in order to check the data for normal distribution. Normal distributed data were compared using the t-Test. For data showing no normal distribution, the Mann-Whitney Rank Sum Test was used. p-values of ≤ 0.05 were considered as statistically significant.

## Figures and Tables

**Figure 1 ijms-20-05426-f001:**
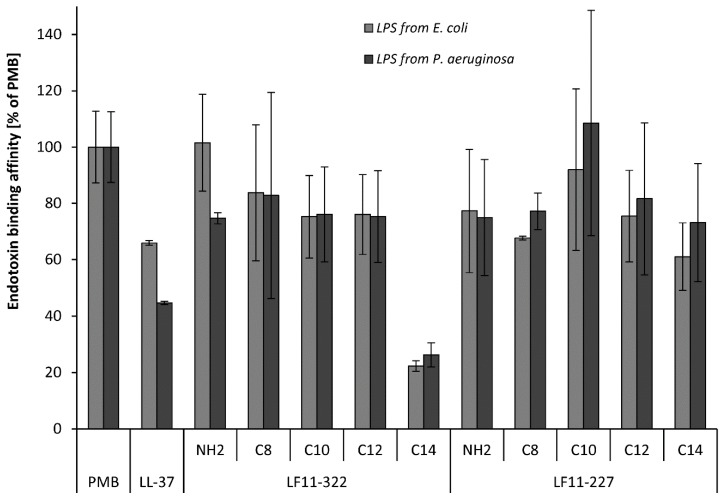
Endotoxin binding affinity of various antimicrobial peptides determined by a fluorescence based displacement method using Bodipy Cadaverine [20] in aqueous solution. Data are given in the percent of the binding affinity of Polymyxin B (*n* = 3).

**Figure 2 ijms-20-05426-f002:**
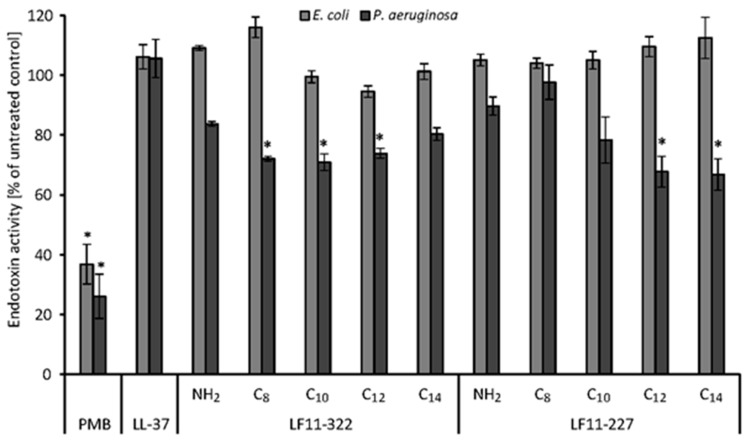
Endotoxin binding activity of 5 ng/mL endotoxin from *E. coli* and *P. aeruginosa* in human Serum. Serum with 5 ng/mL lipopolysaccharides (LPS) and without antimicrobial peptide (AMP) served as positive control and was set to 100%. The endotoxin activity of the peptide spiked serum was related to the positive control. The bars marked with * show a significantly (*p* < 0.05) lower endotoxin activity than the positive control (*n* = 3).

**Figure 3 ijms-20-05426-f003:**
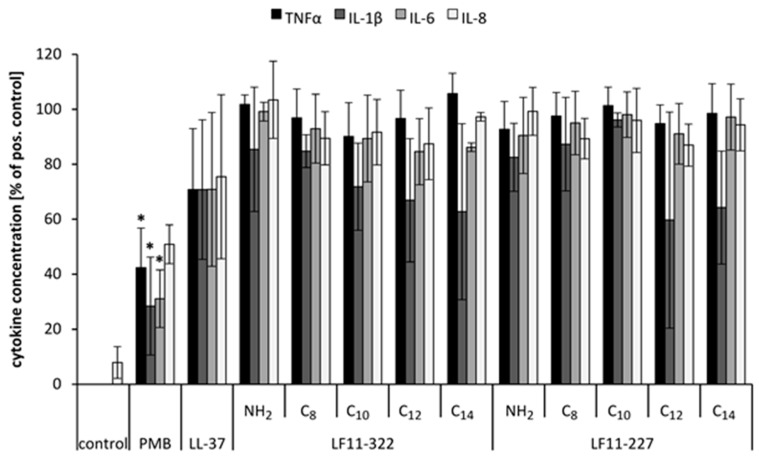
Cytokine levels of endotoxin-spiked blood treated with various AMPs in comparison to the untreated control. Blood without additives served as negative control and blood only with LPS served as positive control. The cytokine level of the positive control was set to 100% and the cytokine concentrations of the samples were compared. The bars marked with * show a significantly (*p* < 0.05) lower cytokine level than the positive control (*n* = 3).

**Figure 4 ijms-20-05426-f004:**
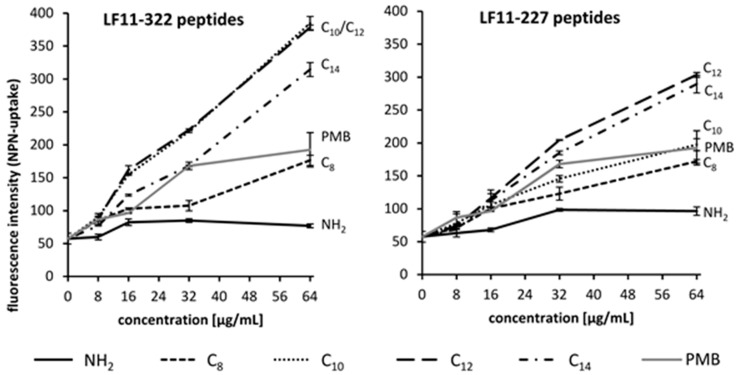
Increasing fluorescence emission after AMP addition by 1-*N*-phenylnaphthylamine (NPN) uptake into the *E. coli* cell. The higher the acylated AMP concentration used, the greater the permeability of NPN was into the bacterial cell (*n* = 3).

**Figure 5 ijms-20-05426-f005:**
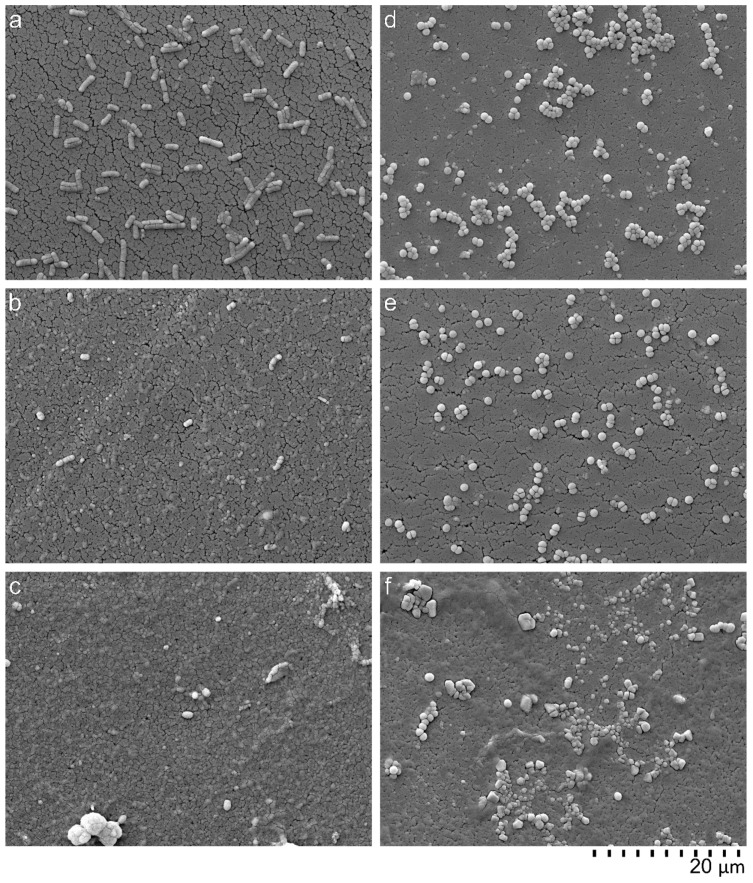
Scanning electron microscope (SEM) images from overnight cultures of *E. coli* (**a**–**c**) and *S. aureus* (**d**–**f**). Bacteria suspensions incubated without peptides (**a**,**d**), incubated with 128 μg/mL PMB (**b**,**e**), and incubated with 128 μg/mL AMP (LF11-322-C_8_) (**c**,**f**).

**Figure 6 ijms-20-05426-f006:**
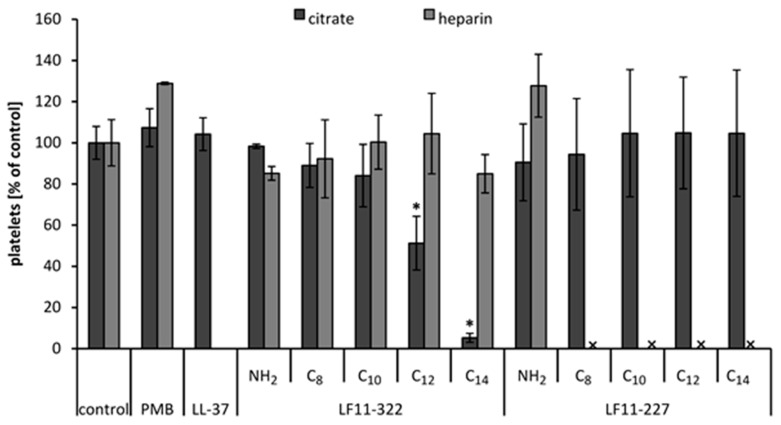
Influence of AMPs on platelets. PRP, anticoagulated with heparin or citrate, was incubated with 128 μg/mL AMP, PMB and LL-37. After 2 h, incubation platelets were counted. PRP without AMPs served as control and was set to 100%. The bars marked with * show a significantly (*p* < 0.05) lower platelet count than the control. x marks the peptides which caused precipitates in heparinized PRP and therefore meant that a platelet count was not feasible (*n* = 3).

**Figure 7 ijms-20-05426-f007:**
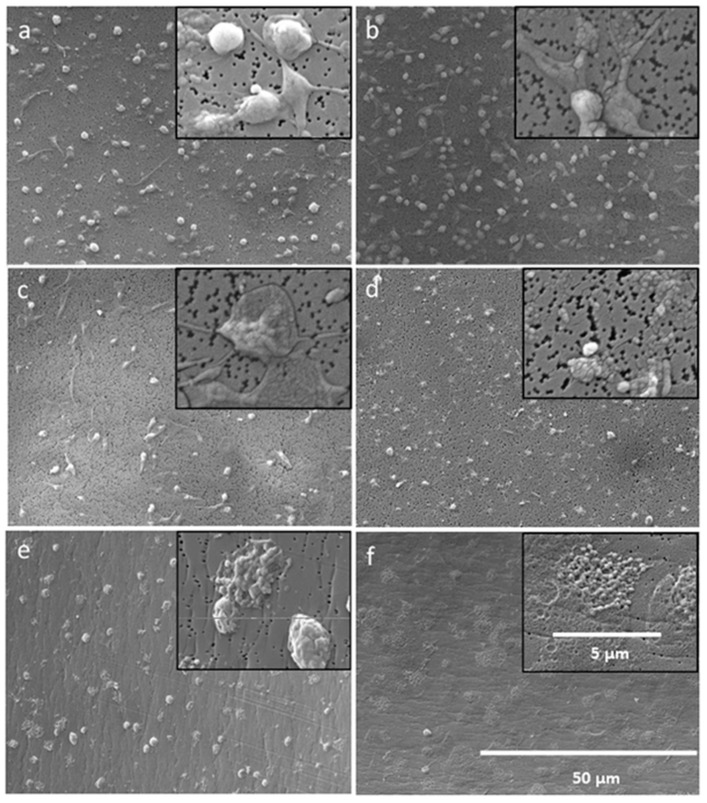
SEM imaging of platelets from citrate anticoagulated PRP incubated without AMPs (**a**), with 128 μg/mL PMB (**b**), with 128 μg/mL 322-NH2 (**c**) and with 128 μg/mL 322-C_14_ (**d**). SEM imaging of platelets from heparin anticoagulated PRP incubated with 128 μg/mL LL-37 (**e**) and with 128 μg/mL LF11-227-C_14_ (**f**).

**Figure 8 ijms-20-05426-f008:**
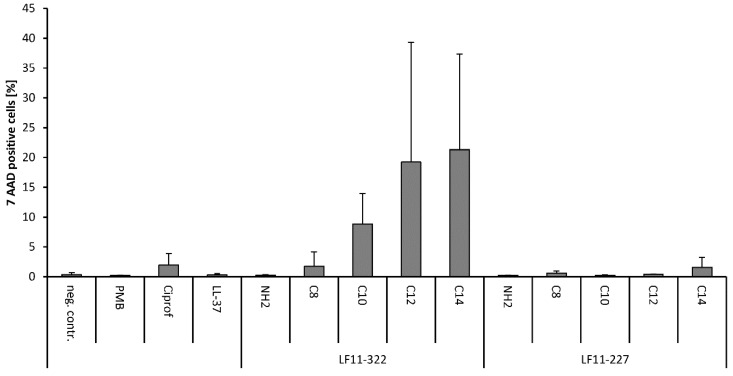
The cytotoxic property of AMPs on human leukocytes was measured by flow cytometry. Whole blood was spiked and incubated with 128 μg/mL AMP, PMB and conventional antibiotics. The apoptotic and dead cells were labelled with the 7-AAD dye and quantified with the flow cytometer (*n* = 3).

**Figure 9 ijms-20-05426-f009:**
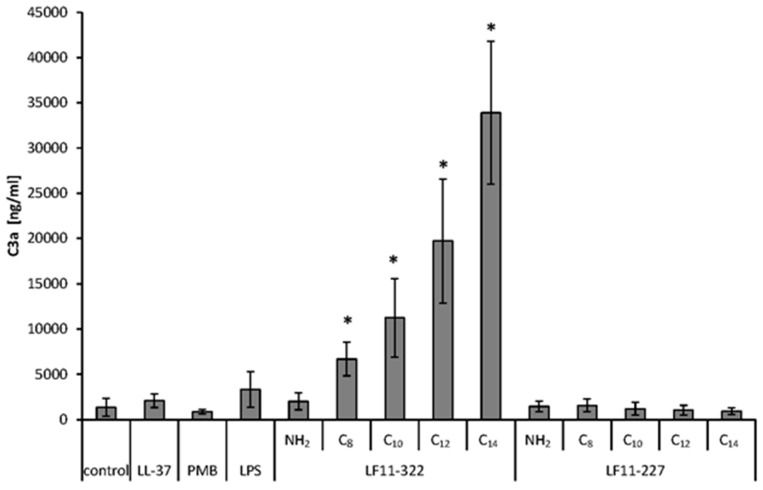
Complement factor C3a activation by AMPs. Citrate anticoagulated plasma was incubated with 128 μg/mL of AMPs, PMB, LL-37, and LPS for 60 min at 37 °C. After incubation, complement factor C3a was quantified via ELISA. Plasma without any additives was used as the control. The bars marked with * show a significantly (*p* < 0.05) higher C3a level than in the negative control (*n* = 3).

**Figure 10 ijms-20-05426-f010:**
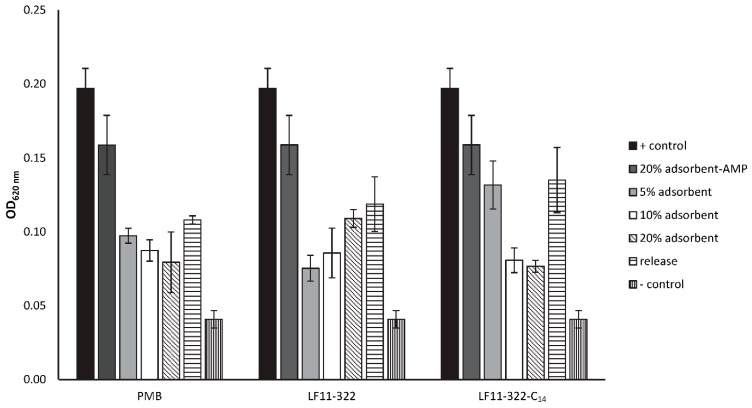
Antimicrobial properties of peptides immobilized on NHS-activated agarose beads against *E. coli*. 5, 10 and 20% (*v/v*) of immobilized adsorbent was incubated with *E. coli* suspension overnight. Bacterial growth was determined by turbidity measurement (OD 620 nm). Bacteria suspension without adsorbent was used as a positive control (+ control). Müller-Hinton broth without bacteria was used as the negative control. Additionally, a bacteria suspension was incubated with adsorbent without immobilized peptides (adsorbent-AMP). To check if the antimicrobial properties caused by a peptide release form the adsorbent, supernatants from 20% adsorbent suspensions were included in the test (release) (*n* = 3).

**Table 1 ijms-20-05426-t001:** Minimum inhibitory concentrations of AMPs, Polymyxin B (PMB), and Ciprofloxacin for *E. coli* and *S. aureus* in aqueous solution, in human serum albumin, and in heparin (5 IU/mL) containing PBS.

		Minimal Inhibitory Concentration (μg/mL)												
	Medium	LF11-322					LF11-227					PMB	LL-37	Ciprofloxacin
		NH_2_	C_8_	C_10_	C_12_	C_14_	NH_2_	C_8_	C_10_	C_12_	C_14_			
*E. coli*	PBS	4	8	8	32	64	8	> 128	> 128	> 128	> 128	2	32	< 0.0625
	4% HSA	16	32	32	32	64	64	128	64	64	64	4	64	< 0.0625
	PBS + heparin	128	> 128	> 128	> 128	> 128	> 128	> 128	> 128	> 128	> 128	4	>128	< 0.0625
*S. aureus*	PBS	4	8	8	16	8	8	> 128	> 128	> 128	> 128	8	> 128	< 0.0625
	4% HSA	> 128	16	16	16	16	> 128	16	16	8	16	64	> 128	< 0.0625
	PBS + heparin	> 128	> 128	> 128	> 128	> 128	> 128	> 128	> 128	> 128	> 128	> 128	>128	< 0.0625

**Table 2 ijms-20-05426-t002:** Influence of AMPs on erythrocyte membranes. Hemolysis was measured in whole blood and in aqueous red blood cell suspension (RBC) at different AMP and PMB concentrations. The free hemoglobin level was determined using the method of Kahn [24] (*n* = 3).

		Free Hemoglobin (mg/dL)					
		BLOOD			RBC in PBS		
peptide conc.		8 (μg/mL)	32 (μg/mL)	128 (μg/mL)	8 (μg/mL)	32 (μg/mL)	128 9 (μg/mL)
LF11-322	NH_2_	5.0 ± 5.3	5.9 ± 4.4	2.6 ± 3.7	10.4 ± 2.4	14.2 ± 3.5	3.2 ± 1.9
	C_8_	2.2 ± 0.5	9.0 ± 3.8	2.7 ± 3.7	12.9 ± 1.5	26.1 ± 2.5	201.9 ± 60.7
	C_10_	2.5 ± 2.0	6.0 ± 4.6	3.0 ± 4.5	16.6 ± 2.4	48.5 ± 5.5	90.1 ± 13.18
	C_12_	2.4 ± 2.6	8.6 ± 8.6	3.2 ± 4.5	15.5 ± 1.2	36.1 ± 2.0	83.2 ± 23.0
	C_14_	2.9 ± 3.4	5.4 ± 3.2	2.9 ± 4.2	12.9 ± 1.2	31.8 ± 3.14	275.0 ± 31.2
LF11-227	NH_2_	2.4 ± 3.1	9.5 ± 12.3	2.5 ± 3.8	10.0 ± 2.2	14.6 ± 3.2	3.2 ± 0.9
	C_8_	3.6 ± 2.8	10.5 ± 13.5	2.3 ± 3.7	13.0 ± 0.7	15.9 ± 2.6	149.8 ± 23.7
	C_10_	2.3 ± 0.6	4.2 ± 1.4	2.7 ± 4.2	18.0 ± 6.7	45.1 ± 20.5	188.0 ± 4.6
	C_12_	2.6 ± 1.4	4.3 ± 2.5	2.5 ± 3.6	11.6 ± 3.2	26.3 ± 12.3	295.2 ± 27.6
	C_14_	3.4 ± 2.1	3.1 ± 0.9	2.6 ± 3.5	18.3 ± 12.8	41.9 ± 19.9	188.5 ± 10.4
PMB		3.5 ± 2.8	3.7 ± 1.4	2.6 ± 3.9	14.9 ± 3.1	18.9 ± 11.2	1.9 ± 0.2
LL-37		1.1 ± 0.7	9.4 ± 13.4	2.6 ± 3.7	2.4 ± 3.6	44.8 ± 17.7	68.4 ± 15.0
control		3.7 ± 1.9			8.5 ± 4.7		

**Table 3 ijms-20-05426-t003:** Influence of AMPs on plasmatic coagulation. AMPs were incubated in citrated and heparinized plasma to determine if they can activate the plasmatic clotting cascade.

Sample		Heparin Plasma	Citrated Plasma
neg. control		−	−
ciprofloxacin		−	−
LL-37		−	−
PMB		−	−
protamine/Ca^++^, Mg^++^		+	+
LF11-322	-NH_2_	−	−
	-C_8_	+	+
	-C_10_	+	+
	-C_12_	+	+
	-C_14_	+	+
LF11-227	-NH_2_	−	−
	-C_8_	+	−
	-C_10_	+	−
	-C_12_	+	−
	-C_14_	+	−

+ plasma clot observed; − no plasma clot observed.

**Table 4 ijms-20-05426-t004:** Influence of acyl chain length of the LF11-322 peptide on plasmatic coagulation.

		Peptide Concentration			
Peptide		8 μg/mL	32 μg/mL	128 μg/mL	256 μg/mL
LL-37		−	−	−	−
PMB		−	−	−	−
LF11-322	-NH_2_	−	−	−	−
	-C_8_	−	−	−	+
	-C_10_	−	−	+	+
	-C_12_	−	+	+	+
	-C_14_	−	+	+	+

− no clotting observed, no D-dimers and normal fibrinogen level; + clotting observed, high D-dimer level, no detectable fibrinogen serum level.

**Table 5 ijms-20-05426-t005:** Coupling efficiency of AMPs in mg per ml adsorbent. PMB and AMPs were coupled on NHS-activated agarose beads. Coupling efficiency was determined using the Pierce 660 nm protein assay.

Peptide	Coupling Efficiency
LF11-322	1.67 ± 0.31 mg/mL
LF11-322-C_14_	0.25 ± 0.09 mg/mL
LF11-227	1.57 ± 0.27 mg/mL
LF11-227-C_14_	0.31 ± 0.11 mg/mL
PMB	1.68 ± 0.23 mg/mL

**Table 6 ijms-20-05426-t006:** AMPs and their amino acid sequence, N-acylation, number of C-atoms in the acylation, purity in % (determined by HPLC) and calculated molecular mass.

Peptide	Amino Acid Sequence	*N*-Acylation	Purity (%)	Calculated MM [Da]
NH_2_-LF11-322	PFWRIRIRR-NH_2_	None	> 99	1298.7
CH_3_(CH_2_)_7_-LF11-322		C_8_ octanoyl-	> 98	1424.9
CH_3_(CH_2_)_9_-LF11-322		C_10_ decanoyl-	> 98	1452.9
CH_3_(CH_2_)_11_-LF11-322		C_12_ lauroyl-	> 99	1480
CH_3_(CH_2_)_13_-LF11-322		C_14_ myristoyl-	> 99	1508
NH_2_-LF11-227	FWRRFWRR-NH_2_	none	> 95	1308.6
CH_3_(CH_2_)_7_-LF11-227		C_8_ octanoyl-	> 96	1434.8
CH_3_(CH_2_)_9_-LF11-227		C_10_ decanoyl-	> 96	1462.9
CH_3_(CH_2_)_11_-LF11-227		C_12_ lauroyl-	> 98	1489.9
CH_3_(CH_2_)_13_-LF11-227		C_14_ myristoyl-	> 98	1517.9

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
