# Peer review of "Blood Compatibility—An Important but Often Forgotten Aspect of the Characterization of Antimicrobial Peptides for Clinical Application"

_ijms, 2019, doi:10.3390/ijms20215426_

Round 1
Reviewer 1 Report
Comments to the authors
The submitted manuscript describes effect of several antimicrobial peptides and their acylated analogs on the various blood components as well as their activity against bacteria.
It is an extensive, proficiently performed experimental work comprising rich experimental repertoire.
In general, I am supportive of this paper, but the manuscript has some issues that need to be corrected before it can be accepted for publication:
The fact that acylation of AMPs increases their antimicrobial (and other) activities is a well- known fact that should be mentioned and quoted somewhere in the text (introduction).
In the introduction section authors should give more information about NH2-LF11-322 and NH2-LF11-227. In the literature LF11 peptide (consisting of 11 amino acid residue) has different amino acid sequence to those sequences given in Table 1. Were the peptides NH2-LF11-322 and NH2-LF11-227 characterized in any previous studies? Please, explain.
Page 2, lines 63-69: The AMPs provided by pba…
I am missing detailed characterization of all peptides shown in Table 1. In which form the peptides were delivered (as for example trifluoroacetates)? What was their HPLC purity?
Table 1 should include the molecular masses of peptides obtained by mass spectrometry measurement together with their theoretical molecular masses.
Page 2, line 71.
Title 2.1. should be: Endotoxin binding affinity in aqueous environment
I am missing any information about the source of LPS. Were these purchased from the company or isolated by authors? Were these LPS provided in the form of powder or as a solution?
Page 3. line 86.
Title 2.2. should be: Endotoxin binding affinity in human plasma
Lines 87-98: Referring to Figure 2 is missing in the text.
Page 4. Lines 100-106
This is a redundant description of what is already described in the text within lines 87-98. Instead of it here should be a title such as Endotoxin binding activity of…in human plasma.
Page 5, lines 117-123
This is already described on page 16, lines 416-422. Discrepancy: 1 ng/ml LPS line 118, on page 16, line 418 endotoxin 0.5 ng/mL.
Page 7. SEM
Which AMP particularly! was used for the experiments related to the cases c and f shown on Figure 6?
Page 8, lines 204-208.
This is already described in experimental part on page 17, lines 466-473. Table 4 needs one sentence title.
Page 9, lines 216-221.
Table 5 needs one sentence title. Description of the experiment is already on page 17, lines 475-487.
Page 10, lines 237-241
Once again, one sentence title for Figure 7 is missing.
Page 11, Figure 8, lines 264-267
…322-NH2 (c) and with…322-C14 (d)
Page 13, lines 299-302
Figure 10. Title of Figure 10 should be like this: Complement factor C3a activation by AMPs
In the Discussion section authors mention the attack of AMPs on the cell membrane of bacteria and erythrocytes. They also open the question of the selectivity of a membrane-destabilizing AMP.
This is the typical mechanism of action ascribed mostly to AMP belonging to the group of amphipathic alpha-helical AMP.
What about NH2-LF11-322 and NH2-LF11-227? Do they belong to the group of alpha-helical peptides? How does the lipofilization alter their structure? I miss any information about the secondary structure of studied AMPs within Discussion section, which in my opinion should be included.
Page 15, line 380
…that PMB selectively destabilizes Gram-negative cell membranes, while Gram-positive..
Page 15, 4. Material and Methods
Here should be included a subchapter titled “Materials and bacterial strains” in which all purchased material will be listed. It should be also described how the peptides were synthesized, purified and how their identity was verified. Also give information how the human plasma was prepared or from which source it was obtained.
Page 16, line 413
Explain briefly what the principle of LAL assay is, and quote it.
Page 16, line 424
The determination of the MIC was carried out in aqueous solution.
Which one? Buffer, medium, or what?
Lines 424-433
The list of material used in this assay should be moved to the subchapter”Material and bacterial strain” and the assay should be described in more detail.
Please, write a brief conclusion, as a separate chapter.
Use italics font for bacterial strains throughout the whole text.
Author Response
Reviewer 1:
|
Issue |
Answer |
|
|
1. The fact that acylation of AMPs increases their antimicrobial (and other) activities is a well- known fact that should be mentioned and quoted somewhere in the text (introduction) |
The following text passage was included in the introduction section: “In order to improve the antimicrobial activity of some AMPs, their hydrophobic properties were increased by targeted N-acylation. This led to increased antimicrobial activity against Escherichia coli, which correlates with the degree of permeability of bacterial membranes through these peptides [3, 4].” |
|
|
2. In the introduction section authors should give more information about NH2-LF11-322 and NH2-LF11-227. In the literature LF11 peptide (consisting of 11 amino acid residue) has different amino acid sequence to those sequences given in Table 1. Were the peptides NH2-LF11-322 and NH2-LF11-227 characterized in any previous studies? Please, explain. |
The following text passages were included in the introduction section:” In this study, LF11-based peptides derived from the pepsin cleavage product of human LF were characterized. Human LF is a multifunctional, iron-binding glycoprotein, which occurs in mammalian exocrine secretions and neutrophil granules. It has antimicrobial and LPS-binding properties[8, 9]. Bovine lactoferricin (LFcin) has a α-helical structure, which is lost in isolated form [10, 11]. Human lactoferricin (hLFcin) comprises amino acid residues 1-45 of the N-terminus of human lactoferrin (hLF) and has increased antibacterial activity compared to intact hLF [12]. LFcin has animicrobial properties against Gram- positive bacteria, Gram- negative bacteria and filamentous yeasts, including some antibiotic-resistant pathogens [13]. Wakabayashi et al.[13, 14] and Strom et al. [15] were able to show that N-acylation of lactoferrin-based peptides could increase their antimicrobial activity. However, this usually leads to higher toxicity to eukaryotic cells, leading to loss of target cell selectivity [16, 17].” “For this study, two synthetic LF11-based AMPs were selected, which showed good antimicrobial and LPS-binding properties in previous studies [18, 19].” |
|
|
3. Page 2, lines 63-69: The AMPs provided by pba… I am missing detailed characterization of all peptides shown in Table 1. In which form the peptides were delivered (as for example trifluoroacetates)? What was their HPLC purity? Table 1 should include the molecular masses of peptides obtained by mass spectrometry measurement together with their theoretical molecular masses. |
The following text passage was included in the introduction section:” The AMPs were obtained in lyophilized form and dissolved in 0.1 M 2-(N-Morpholino)ethansulfonacid (MES) buffer at a concentration of 1 mg/ml. The peptide solutions were prepared weekly and stored at 4 °C." The table was extended. Degree of purity and calculated peptide mass were added. Recording to the suggestion of an other reviewer we moved the text and the table to the material and method section. |
|
|
4. Page 2, line 71. Title 2.1. should be: Endotoxin binding affinity in aqueous environment I am missing any information about the source of LPS. Were these purchased from the company or isolated by authors? Were these LPS provided in the form of powder or as a solution? |
Heading 2.1 was amended.
The following text passage was included in the Material and Methods section: “LPS from E. coli 055:B5 and P. aeruginosa 10 as well as PMB and the human host defense peptide LL-37 (LL-37 trifluoroacetate salt) were purchased from Sigma-Aldrich (St. Louis, Missouri, USA). Ciprofloxacin and Vancomycin were purchased from Sigma-Aldrich (St. Louis, MO, USA).”
|
|
|
5. Page 3. line 86. Title 2.2. should be: Endotoxin binding affinity in human plasma |
Heading 2.2 was amended.
|
|
|
6. Lines 87-98: Referring to Figure 2 is missing in the text |
Referring to Figure 2 is now included in the text. |
|
|
7. Page 4. Lines 100-106 This is a redundant description of what is already described in the text within lines 87-98. Instead of it here should be a title such as Endotoxin binding activity of…in human plasma. |
The Figure legend was changed to: “Endotoxin binding activity of 5 ng/ml endotoxin from E. coli and P. aeruginosa in human Serum. Serum with 5 ng/mL LPS without AMP served as positive control and was set to 100 %. The endotoxin activity of the peptide spiked serum was related to the positive control. The bars marked with * show a significantly (p<0.05) lower endotoxin activity than the positive control.” |
|
|
8. Page 5, lines 117-123 This is already described on page 16, lines 416-422. Discrepancy: 1 ng/ml LPS line 118, on page 16, line 418 endotoxin 0.5 ng/mL. |
The Figure legend was changed to: “Cytokine levels of endotoxin spiked blood treated with various AMPs in comparison to the untreated control. Blood without additives served as negative control and blood only with LPS served as positive control. The cytokine level of the positive control was set to 100 % and the cytokine concentrations of the samples were compared. The bars marked with * show a significantly (p<0.05) lower cytokine level than the positive control.” The discrepancy was corrected.
|
|
|
9. Page 7. SEM Which AMP particularly! was used for the experiments related to the cases c and f shown on Figure 6? |
For this experiment the AMP LF11-322-C8 was chosen. We have included it into the figure legend. |
|
|
10. Page 8, lines 204-208. This is already described in experimental part on page 17, lines 466-473. Table 4 needs one sentence title. |
In the course of the corrections in the document, the numbering of the tables has changed. The former table 4 is now table 3.
The table 3 legend has been changed to: “Influence of AMPs on plasmatic coagulation. AMPs were incubated in citrated and heparinized plasma to determine if they can activate the plasmatic clotting cascade. + ... plasma clot observed; - ... no plasma clot observed.” |
|
|
11. Page 9, lines 216-221. Table 5 needs one sentence title. Description of the experiment is already on page 17, lines 475-487. |
In the course of the corrections in the document, the numbering of the tables has changed. The former table 5 is now table 4.
The table 4 legend has been changed to: “Influence of acyl length of the LF11-322 peptide on plasmatic coagulation. - ... no clotting observed, no D-dimers and normal fibrinogen level; + ... clotting observed, high D-dimer level, no detectable fibrinogen serum level.” |
|
|
12. Page 10, lines 237-241 Once again, one sentence title for Figure 7 is missing. |
Title of figure 7 was added: “Influence of AMPs on platelets.” |
|
|
13. Page 11, Figure 8, lines 264-267 …322-NH2 (c) and with…322-C14 (d) |
Missing label (c) and (d) was inserted. |
|
|
14. Page 13, lines 299-302 Figure 10. Title of Figure 10 should be like this: Complement factor C3a activation by AMPs |
Suggested figure title was added in the figure legend. |
|
|
15. In the Discussion section authors mention the attack of AMPs on the cell membrane of bacteria and erythrocytes. They also open the question of the selectivity of a membrane-destabilizing AMP. This is the typical mechanism of action ascribed mostly to AMP belonging to the group of amphipathic alpha-helical AMP. What about NH2-LF11-322 and NH2-LF11-227? Do they belong to the group of alpha-helical peptides? How does the lipofilization alter their structure? I miss any information about the secondary structure of studied AMPs within Discussion section, which in my opinion should be included. |
The following text passage was included in the discussion section:” In the presence of a membrane mimetic environment, the short (nine amino-acids) LF11-322 folds into a loop comprising a short helical segment [39], which separates the cationic and hydrophobic residues along the molecular axis of the peptide. The secondary structure prediction using the program PEP-FOLD indicated a similar structure of LF11-227 (http://bioserv.rpbs.univ-paris-diderot.fr/services/PEP-FOLD/). The structure of both peptides is in accordance with the structure of the parent peptide LF11. N-acylation forced the peptide chain to wrap around the acyl chain resulting in an even better defined fold [40]. It can be expected that N-acylation of LF11-322 and LF11-227 follows the same principle.” |
|
|
16. Page 15, line 380 …that PMB selectively destabilizes Gram-negative cell membranes, while Gram-positive..
|
This error was corrected.
|
|
|
17. Page 15, 4. Material and Methods Here should be included a subchapter titled “Materials and bacterial strains” in which all purchased material will be listed. It should be also described how the peptides were synthesized, purified and how their identity was verified. Also give information how the human plasma was prepared or from which source it was obtained. |
The subchapter “Materials” and “Blood donation” were included. Following text passage was added to the Material section: “Peptides were purchased from PolyPeptide Laboratories (San Diego, CA, USA) and were synthesized using FMOC-chemistry. Purity of the peptides were >96% as determined by RP-HPLC and MS.” |
|
|
18. Page 16, line 413 Explain briefly what the principle of LAL assay is, and quote it. |
Following text passage was included in the material section: “The LAL test is the most established method for quantifying LPS. This method uses amebocyte lysate from Limulus polyphemus, wherein the presence of LPS triggers a defense mechanism that induces coagulation. The LAL test is used successfully for detection of LPS in a wide variety of industrial, pharmaceutical, and research applications, such as water supplies, parenteral fluids, drugs for intravenous administration, and certain biologic fluids (e.g., cerebrospinal fluid).” |
|
|
19. Page 16, line 424 The determination of the MIC was carried out in aqueous solution. Which one? Buffer, medium, or what? |
The wording “aqueous solution” was more specified in more detail: “The determination of the MIC was carried out in aqueous solution (0.01 M phosphate buffer, 0.0027 M potassium chloride and 0.137 M sodium chloride, pH 7.4) as well as …” |
|
|
20. lines 424-433 The list of material used in this assay should be moved to the subchapter ”Material and bacterial strain” and the assay should be described in more detail. |
These suggested revisions were implemented. |
|
|
21. Please, write a brief conclusion, as a separate chapter. |
A brief conclusion was added as separate chapter. |
|
|
22. Use italics font for bacterial strains throughout the whole text. |
This change was made for the entire manuscript. |
|
Reviewer 2 Report
The manuscript deals with a very interesting and crucial matter regarding the properties of antimicrobial peptides and how can we assess their efficacy in physiological conditions. The authors put forward an important issue: how the binding of AMPs to heparin can alter the blood coagulation pathway. However, the results presentation and interpretation is not sufficient to support the conclusions.
Before acceptance, the manuscript needs a major revision.
Main issues:
I think that an experiment where the peptide interaction with heparin is directly measured would be compulsory to support the paper conclusions.
Most results are interpreted taking too many parameters simultaneously. How can we know what is attributed to micelles formation, BSA, heparin or LPS binding?
Abstract: line 22: please modify statement to indicate clearly that interaction is with alkylated peptides.
Pdf version has an error and references to tables and figures are no correctly visualized.
Please check end of introduction and starting of results section:
Lines 63-67 are not matching with an end of introduction. Also, table 1 should go to results.
Before section 2.1, it would be necessary to include an introductory section describing the peptides and the rationale to design them.
Figure 1 legend should be self- explanatory. Indicate that E. coli and P.aeruginosa refer to LPS.
Lines 128-129: This is probably due to… (this is more a statement proper to discussion than to results section)
Lines 131-137: too many parameters are considered here simultaneously. How can we know what is attributed to BSA, heparin or LPS binding?
MIC is evaluated in the presence of heparin, but it would be necessary to know the direct affinity of the peptides to heparin.
Line 139: “the reason might be”: this is a statement for discussion, not for results.
Line 141: when introducing ciprofloxacin, it is important to explain which kind of antibiotic it is, and which is the working mechanism. Why has it been selected as a control and references to previous work. Previous studies related to heparin binding?
Figure 4 is only an hypothesis, more appropriate to add at the end of the manuscript. Also, revise figure 4 legend to indicate clearly in what the illustration is based.
In addition, it is important to add a reference in the text (better in the discussion section) of previous structural work on peptide binding to heparin.
Section 2.5: start with a sentence explaining how the methodology works (even if the method is detailed in the methodology section, here it is important to explain what we are measuring by NPN uptake.
Lines 161-162: explain why the observed differences between C12 and C14 peptide. Relation to micelle formation?
SEM is not the best methodology to assess membrane lysis. It would be necessary to use TEM here.
Besides the amplification is not enough to show the bacteria wall damage.
Line 190: no direct assay of peptide-BSA binding is provided.
Table 4: please add a header to the table.
In table 4 another antibiotic (vancomycin) is added as a control. This needs to be explained in the text and justified. Likewise with the use of protamine.
Table 5: add a header.
Line 256: it is unclear how and by which methodology turbidity is observed
Figure 8: information for c and d is missing.
Table 6: coupling efficiency should preferably be calculated from more than one experiment. Otherwise, it might be better to put this table in the supplemental.
Lines 329-331: is it always necessary to have an alkylated AMP to achieve endotoxin neutralization?
Lines 337-338: it is important here to give a plausible explanation why differences are observed between the two types of LPS.
Lines 348-349: again, figure 4 is not a direct explanation
Discussion text is confusing. Particularly the last section. Also, no conclusions are provided.
Line 412: provide commercial source for LPS
Line 425. Indicate standardization…
Line 456: indicate source of blood: hospital,…
Line 468: indicate source of plasma
Figure 12 can be shifted to the supplemental section
Author Response
Reviewer 2:
|
Issue |
Answer |
|
|
1. line 22: please modify statement to indicate clearly that interaction is with alkylated peptides. |
This proposed amendment was implemented. |
|
|
2. Pdf version has an error and references to tables and figures are no correctly visualized |
This error was based on the conversion of the online submission system and should be correct now. |
|
|
3. Before section 2.1, it would be necessary to include an introductory section describing the peptides and the rationale to design them. |
The suggestion of the other reviewers led to the inclusion of a separate subsection "Antimicrobial Peptides" in material and methods. There the peptides as well as their synthesis are described in more detail. The table has also been moved to the Material and Methods section. In addition, the was extended with purity and molecular mass of each peptide. |
|
|
4. Figure 1 legend should be self- explanatory. Indicate that E. coli and P. aeruginosa refer to LPS. |
Figure legend was changed into: LPS from E. coli and LPS from P. aeruginosa. |
|
|
5. Lines 128-129: This is probably due to… (this is more a statement proper to discussion than to results section) |
This text passage was removed from the result part. |
|
|
6. Lines 131-137: too many parameters are considered here simultaneously. How can we know what is attributed to BSA, heparin or LPS binding? |
The authors think that this feedback is based on a misunderstanding. The aims of this part of the paper was to test the antimicrobial effect of AMPs in different media by using the MIC test (not the LAL test). No LPS was used here. Another aim was to show the effect of heparin, namely the reduction of the antimicrobial effect in PBS vs PBS+heparin. |
|
|
7. MIC is evaluated in the presence of heparin, but it would be necessary to know the direct affinity of the peptides to heparin. |
In this study only heparin was observed to have an effect on the antimicrobial properties of LF11 peptides. A direct affinity test was not included in this study. A section which discusses this aspect (incl. citations) was added to the discussion. See also comment to feedback no 11. |
|
|
8. Line 139: “the reason might be”: this is a statement for discussion, not for results. |
This part was moved to the discussion. |
|
|
9. Line 141: when introducing ciprofloxacin, it is important to explain which kind of antibiotic it is, and which is the working mechanism. Why has it been selected as a control and references to previous work. Previous studies related to heparin binding? |
Following text with references were included: “Ciprofloxacin was used as a control antibiotic because it has a broad spectrum of action at very low dosage. The MIC90 against all species of Enterobacteriaceae and Staphylococcus is < 1 mg/L [21]. In past studies, no mutual influences between ciprofloxacin and heparin could be observed [22].” |
|
|
10. Figure 4 is only an hypothesis, more appropriate to add at the end of the manuscript. Also, revise figure 4 legend to indicate clearly in what the illustration is based. |
As suggested from another reviewer, figure 4 was removed from the manuscript. |
|
|
11. In addition, it is important to add a reference in the text (better in the discussion section) of previous structural work on peptide binding to heparin. |
Following text with references were included in the discussion part: “Previous work has shown interactions between heparin and natural cationic AMPs such as LL-37 and protamine. These studies also assume an ionic interaction between the anionic heparin and the cationic peptides [32, 33].” |
|
|
12. Section 2.5: start with a sentence explaining how the methodology works (even if the method is detailed in the methodology section, here it is important to explain what we are measuring by NPN uptake. |
The capture 2.5 starts now with the explaining of the NPN uptake assay: “To determine whether the AMPs can permeabilize the outer membrane of Gram-negative bacteria, the AMP induced 1-N-Phenylnaphtylamine (NPN) uptake was determined. NPN is a hydrophobic fluorescent dye whose fluorescence emission is improved in an environment of glycerophospholipid [23]. A rising fluorescence signal therefore means a destabilization, destruction or loss of the outer membrane of Gram-negative bacteria.” |
|
|
13. Lines 161-162: explain why the observed differences between C12 and C14 peptide. Relation to micelle formation? |
In chemical surfactants, it has been shown that the longer the lipophilic acyl chain, the more effective the micellization, since the intermolecular hydrophobic interactions increase (29). Previous studies (38) show that the micellar concentration of hexadecanoic acid conjugated to an amino acid is in the micromolar range and should decrease significantly by increasing the peptide chain. This indicates that lipopeptides with long fatty acids exist at least partially as micelles in the concentrations in which they perform a biological function. |
|
|
14. SEM is not the best methodology to assess membrane lysis. It would be necessary to use TEM here. |
The authors agree that TEM, due to its higher resolution, would be a good tool to assess membrane lysis. However, the goal of the REM methodology was to show potential differences in the morphology of the whole bacteria culture which could indicate a change in membrane morphology. The corresponding figure was updated by removing the high resolution inserts. With the REM available at our institute it is not possible to characterize membrane substructures. Most importantly, the REM methodology was used to support the other methods used in this paper and not as the only method to characterize the impact of AMPs on bacteria. |
|
|
15. Besides the amplification is not enough to show the bacteria wall damage. |
See comment to remark no 14. |
|
|
16. Line 190: no direct assay of peptide-BSA binding is provided |
The explanation for reduced hemolysis in plasma vs RBC was formulated more cautiously. |
|
|
17. Table 4: please add a header to the table. |
A header for table 4 was added.
(Please consider that in the course of the corrections in the document, the numbering of the tables has changed.) |
|
|
18. In table 4 another antibiotic (vancomycin) is added as a control. This needs to be explained in the text and justified. Likewise with the use of protamine. |
Vancomycin was used as a control antibiotic since it is commonly used for intravenous applications in the clinic. However, the authors agree that the results for vancomycin in this paper do not substantially contribute to the general understanding of the manuscript. Therefore, it was removed from the manuscript. |
|
|
20. Table 5: add a header. |
A header for table 5 was added. |
|
|
21. Line 256: it is unclear how and by which methodology turbidity is observed |
The turbidity was caused by plasmatic coagulation. The text was rewritten more clearly. |
|
|
22. Figure 8: information for c and d is missing. |
Missing labels (c) and (d) were inserted. |
|
|
23. Table 6: coupling efficiency should preferably be calculated from more than one experiment. Otherwise, it might be better to put this table in the supplemental. |
Coupling efficiency was calculated from 3 experiments. Standard deviations were added. |
|
|
24. Lines 329-331: is it always necessary to have an alkylated AMP to achieve endotoxin neutralization? |
Probably not. Therefore, the corresponding part of the discussion was changed into a more cautious wording: “Studies from other research groups have shown that for certain cationic AMPs, a fatty acid substitution of ≥ C8 is necessary for efficient LPS neutralization [29,30]” |
|
|
25. Lines 337-338: it is important here to give a plausible explanation why differences are observed between the two types of LPS. |
The following explanation was added: “This can be attributed to the fact that bacterial strains differ in the acylation (number, length) of the lipid A moiety of their endotoxins.” |
|
|
26. Lines 348-349: again, figure 4 is not a direct explanation |
Figure 4 was removed from the manuscript. |
|
|
27. Discussion text is confusing. Particularly the last section. Also, no conclusions are provided. |
A short conclusion was added. |
|
|
28. Line 412: provide commercial source for LPS |
Has been included in the Material and Method section. |
|
|
29. Line 425. Indicate standardization… |
The text was adapted in order to indicate that the procedure is based on a standardized method. |
|
|
30. Line 456: indicate source of blood: hospital,… |
Was included in the Material and Method section. |
|
|
31. Line 468: indicate source of plasma |
Was included in the Material and Method section. |
|
|
32. Figure 12 can be shifted to the supplemental section |
Figure 12 was removed from the manuscript. |
|
Reviewer 3 Report
An interesting set of points are brought up by this study, regarding the role of hydrophobicity in relation to the effects of factors in the blood on the activity of antimicrobial peptides. Some issues to address:
Page 2, lines 63-67. This does not appear to belong in the introduction. It should be moved to the materials and methods section. Figure 4. This is an unnecessary and confusing figure. In the text it is very clear that the negatively charged heparin could effectively neutralize and inactivate the positively charged peptides. The figure as shown implies a double-helical structure, which has not been determined by experimentation. Page 7, line 179. Which AMP was used in this figure? Page 10, line 249, please correct the spelling of hydrophobic. Page 11, line 265, what is figure 8 c and d? Page 12, lines 290-291. Please change “way” to “pathway” or “pathways” as appropriate. Page 14, line 322 (and elsewhere in the manuscript). Please change “bouillon” to “broth”. Page 15, line 348. As correctly pointed out in the results section, heparin “might” inactivate the peptide by charge neutralization, but this has not been shown experimentally. Page 15, line 385, please change “contains a lot of Arginine” to “are arginine-rich”. Page 19, line 560, change to “statistics”.
Author Response
Reviewer 3:
|
Issue |
Answer |
|
|
1. Page 2, lines 63-67. This does not appear to belong in the introduction. It should be moved to the materials and methods section |
This proposed amendment was implemented. |
|
|
2. Figure 4. This is an unnecessary and confusing figure. In the text it is very clear that the negatively charged heparin could effectively neutralize and inactivate the positively charged peptides. The figure as shown implies a double-helical structure, which has not been determined by experimentation. |
Figure 4 was removed from the manuscript. |
|
|
3. Page 7, line 179. Which AMP was used in this figure? |
The AMP used for this image is now specified in the figure legend. |
|
|
4. Page 10, line 249, please correct the spelling of hydrophobic. |
The typo has been corrected. |
|
|
5. Page 11, line 265, what is figure 8 c and d? |
Missing label (c) and (d) was inserted. |
|
|
6. Page 12, lines 290-291. Please change “way” to “pathway” or “pathways” as appropriate. |
This proposed amendment was implemented. |
|
|
7. Page 14, line 322 (and elsewhere in the manuscript). Please change “bouillon” to “broth”. |
These suggested changes were made. |
|
|
8. Page 15, line 348. As correctly pointed out in the results section, heparin “might” inactivate the peptide by charge neutralization, but this has not been shown experimentally. |
This text passage was changed into: “The polyanionic heparin is able to cancel the antimicrobial effect of the AMPs in our MIC study. A possible explanation for this could be that the anionic heparin binds the cationic AMPs and thus inactivates them. Previous works have shown interactions between heparin and natural cationic AMPs such as LL-37 and protamine [33, 34]. These studies also assume an ionic interaction between the anionic heparin and the cationic peptides. Further and especially targeted studies would be necessary to clarify this observed effect.” |
|
|
9. Page 15, line 385, please change “contains a lot of Arginine” to “are arginine-rich”. |
This suggested change was made. |
|
|
10. Page 19, line 560, change to “statistics” |
This suggested change was made. |
|
Round 2
Reviewer 2 Report
The authors have addressed most of the raised concerns. before final acceptance I would only introduce few editing changes:
line 22: better indicate: "by the presence of heparin"
lines 79-81: details about purity or source of AMPs should go in the methodology section. They can be included also in tabe 1 footnote, but not at the end of the introduction.
lines 142-143: sentence "in order to determine whether animcirobial activity is afected by heparin..." should be clearly highlighted previously as one of the objectives of the manuscript.
line 175: change sentence: "SEM was used to visualize killing of bactria cells by lysing..." This is not correct. We can only infer cell wall damage by SEM.
Author Response
Reviewer 2:
|
Issue |
Answer |
|
1. line 22: better indicate: "by the presence of heparin" |
The sentence was adapted |
|
2. lines 79-81: details about purity or source of AMPs should go in the methodology section. They can be included also in tabe 1 footnote, but not at the end of the introduction.
|
The suggested change was made. |
|
3. lines 142-143: sentence "in order to determine whether animcirobial activity is afected by heparin..." should be clearly highlighted previously as one of the objectives of the manuscript.
|
Following text was included into the introduction (aims): ‘”Furthermore, it was examined whether the strongly polyanionic heparin has an influence on the antimicrobial effect of cationic AMPs.” |
|
4. line 175: change sentence: "SEM was used to visualize killing of bacteria cells by lysing..." This is not correct. We can only infer cell wall damage by SEM.
|
The suggested change was made: “SEM was used to visualize cell wall damage of bacteria cells.” |